# Modeling Multi-Scale Scientific Impact via Heterogeneous Networks and LLMs

## Abstract

Accurately estimating the future impact of scientific work is essential for understanding research dynamics and guiding funding, hiring, and policy decisions. Despite growing interest, two challenges still remain unresolved: (i) the heterogeneous, multi-scale factors of scientific impact, spanning short-term citations to long-term disciplinary influence, and (ii) current methods fail to fully capture domain-specific expertise, which could be better leveraged through the rich knowledge embedded in large language models (LLMs). To address these challenges, we introduce a unified framework that couples heterogeneous graph neural networks with frozen LLM backbones to predict scientific impact, specifically citation counts at yearly and monthly horizons. Our method trains only lightweight prefix-tuning token embeddings from graph features, enabling scalable learning while retaining strong semantic representations. To enable comprehensive training and evaluation, we provide a large-scale and multi-temporal benchmark with rich metadata, multiple impact indicators, and accompanying tools for future research. Experimental results show that our approach consistently surpasses both traditional and LLM-only baselines, reducing error by over 25% for yearly prediction and 18% for monthly prediction. Interestingly, we also find that directly fine-tuning LLMs on this task can induce bias, whereas using their hidden representations as enhanced features yields superior performance. All datasets, tools and code will be released on GitHub.

## 1 Introduction

Understanding and predicting the impact of scientific research has long been a central pursuit in the science of science literature (Bai et al., 2020b; Xu et al., 2022; Xia et al., 2023). Accurate modeling of scientific impact is not only of theoretical interest. It also provides the foundation for many practical applications, such as identifying high-potential research, optimizing funding allocation, and informing institutional evaluation (Zhang & Wu, 2024). Despite its importance, reliable prediction of scientific influence remains an open challenge. Scientific impact is inherently complex because it is simultaneously *multi-scale*, *heterogeneous*, and *dynamic* (Wu et al., 2019; Iqbal et al., 2020; Giovanni et al., 2020; Xu et al., 2022; Dong et al., 2016; Gebhart & Funk, 2023).

Impact may appear immediately through early citations or social media attention, but it may also unfold slowly over decades, shaping entire disciplines and even affecting non-academic domains (Tahamtan & Bornmann, 2020; Abramo et al., 2024). Predictive models must therefore account for signals at multiple timescales and accommodate diverse drivers of influence, ranging from scientific factors such as novelty, topicality, and methodological rigor to non-scientific factors such as author reputation, affiliation, and domain-specific citation practices (Xing et al., 2024; Zhao et al., 2025).

To capture multiple factors, early work modeled impact with handcrafted features and conventional classifiers, leveraging signals such as first-year citations, author h-index, and venue prestige (Ibáñez et al., 2009; Yang & Han, 2023), but these approaches fall short in capturing the nonlinear, context-dependent dynamics of influence. Subsequent research introduced temporal models (Wang et al., 2013), citation-trajectory analyses (Jiang et al., 2021), and deep neural networks that learn sequential growth patterns (He et al., 2023b). More recently, graph neural networks (GNNs) have advanced the field by capturing interdependencies among authors, venues, affiliations, and papers, underscoring the value of structural signals for long-term impact (Xue et al., 2024). Given the strong performance

of large language models (LLMs) across diverse tasks, a promising next step is to fuse graph-based structure with the semantic representations learned by LLMs.

Domain-adapted pre-trained models such as SciBERT (Beltagy et al., 2019a) and SPECTER (Cohan et al., 2020) capture semantic cues related to novelty, interdisciplinarity, and methodological rigor. Recent studies further show that LLMs can forecast long-term scientific impact directly from text, without relying on citation or network information (Zhao et al., 2025; de Winter, 2024). Overall, these findings highlight the predictive power of textual content, yet also reveal a persistent gap: most approaches emphasize either structural dependencies or textual semantics in isolation, and only a few explore their integration across temporal scales.

However, this lack of integration limits progress towards a comprehensive modeling of scientific impact. Predictive frameworks should combine the semantic richness of textual representations with the structural and temporal context captured by scholarly networks. By doing so can we capture the full spectrum of factors that shape scientific influence. Therefore, in this work, we aim to close this gap by addressing two research questions: **RQ1:** What are effective strategies for incorporating LLMs into heterogeneous network-based models of multi-scale scientific impact? **RQ2:** How do scientific factors (e.g., content, topic, novelty) and non-scientific factors (e.g., author reputation, institutional affiliation, venue prestige) jointly influence the prediction of research impact?

To address these questions, we present a unified framework that integrates heterogeneous graph neural networks with pretrained LLMs. As a first step, we examine applying LLMs directly to temporal sequence prediction and prove that they are prone to exploiting dataset-level numeric priors rather than modeling substantive scientific factors. Motivated by this, we introduce a prefix-tuning strategy that uses LLMs as feature extractors to encode research topics and domain knowledge. The resulting model captures both structural dependencies in scholarly knowledge graphs and the semantic information contained in research texts.

To enable comprehensive evaluation, we constructed a million-scale benchmark that integrates metadata, multiple impact indicators, and citation graphs spanning ten disciplines. In addition, our benchmark supports automated subset extraction via reusable tools, incorporates denoising pipelines and includes multimodal content. In summary, the contributions are as follows:

- **Large-scale multi-temporal benchmark and tools.** We construct a million-scale benchmark spanning ten disciplines that integrates rich metadata, multiple impact indicators, and citation graphs to support short- and long-term training and evaluation.

- **Unified structure–semantics–time modeling.** We propose a unified framework that fuses heterogeneous scholarly graphs with LLM-derived textual representations to jointly model author–institution–venue–paper dependencies, semantic signals, and temporal dynamics.

- **Analysis and empirical insights.** We disentangle and quantify the roles of scientific and non-scientific factors—such as topical novelty, interdisciplinarity, and reputational signals—in shaping impact, yielding actionable insights into scholarly dynamics.

## 2 RELATED WORK

**Predicting Scientific Impact.** Evaluating and predicting scientific impact has long relied on citation counts as a proxy for influence (Fu & Aliferis, 2008; Vincent & Yves, 2009; Bai et al., 2020a). Traditional bibliometrics like the h-index reflect established rather than future impact, prompting interest in article impact prediction based on early signals (Yang & Han, 2023; Lahatte & Turckheim, 2024; Vital Jr et al., 2024). Initial methods used handcrafted features—author reputation, venue rank, early citations—with regression or classification models, but struggled to capture the complexity of impact dynamics (Ibáñez et al., 2009). More recent approaches adopt data-driven and network-aware models, including temporal models that simulate citation growth via paper "fitness" and decay (Wang et al., 2013; Tonta & Akbulut, 2019; Correa et al., 2020). Jiang et al. (2021) proposed HINTS, an end-to-end model predicting citation time series from publication time using pre-publication metadata and bibliographic networks, effectively addressing the cold-start problem and outperforming models dependent on years of citation data. Xue et al. (2024) introduced a GNN-based framework leveraging dynamic citation graphs and auxiliary tasks to yield interpretable and accurate predictions across paper lifespans and disciplines. These developments highlight the value of modeling scientific impact as a multifactorial process rather than a singular metric.

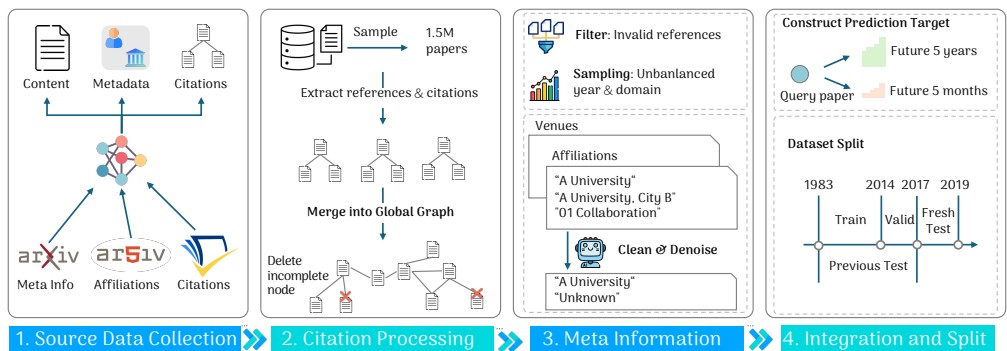

Figure 1: The construction pipeline of our dataset, including four main steps: (1) Crawl raw data from three sources; (2) extract sampled papers and citation graphs; (3) balance the dataset distribution and clean noisy fields; and (4) collect multi-grained labels and perform dataset split.

**Heterogeneous Scholarly Networks** Scholarly impact features are effectively modeled as heterogeneous information networks (HIN), where nodes (e.g., papers, authors, venues) and edges (e.g., citations, co-authorship) represent diverse entities and relations (Geng et al., 2022; He et al., 2023a; Abramo et al., 2021). Meta-path analysis enabled early relation-specific influence measures (Sun et al., 2011; Zhao et al., 2024a), followed by ranking methods that integrated content, venue, and publication networks (Tang et al., 2008). More recent work uses temporal GNNs and embedding alignment to track evolving influence and predict impact by embedding papers into historical network contexts Saier et al. (2021); Hirako et al. (2023); Holm et al. (2021). Advanced models apply relational GNNs with attention and temporal encoders to weigh author, venue, and content signals differently over time. HIN-based methods have also addressed collaboration prediction, topic emergence, and prestige estimation, confirming their strength for multi-scale scientific impact analysis Zhao et al. (2024b); de Winter (2024); Arts et al. (2024); Jin et al. (2024). These models capture network structure and temporal context but underutilize the semantic richness of texts.

**LLMs for Scientific Understanding.** With the fast development of LLMs, researchers are increasingly exploring their ability to predict scientific impact from textual content alone Cohan et al. (2020); Geng et al. (2023); He et al. (2023a). Early efforts used simple representations like TF-IDF, but newer models leverage transformers trained on scientific corpora. SciBERT (Beltagy et al., 2019b) improved classification and recommendation by capturing domain-specific language, while SPECTER (Cohan et al., 2020) showed that content-based embeddings correlate with scholarly relevance. Zhao et al. (2025) proposed "From Words to Worth," a content-only LLM framework predicting impact from titles and abstracts in a double-blind fashion, achieving state-of-the-art performance using a field- and time-normalized metric, $TNCSI_{SP}$. Similarly, Vital Jr et al. (2024) showed GPT-based abstract embeddings could effectively identify highly cited papers, and even TF-IDF performed competitively, underscoring the role of topical relevance. de Winter (2024) found ChatGPT-4's qualitative scores on novelty, clarity, and engagement significantly correlated with later citations and Altmetric scores, suggesting LLMs can evaluate intangible manuscript qualities linked to impact. These studies show that LLMs excel at capturing intrinsic semantic merit and can complement or even rival traditional metadata-driven models. Combining LLM-derived content features with graph-based signals promises a holistic, multi-scale approach to modeling scientific influence.

## 3 DATASET CONSTRUCTION

To facilitate systematic research on scientific impact prediction, we introduce a new large-scale dataset together with open-source tools for automated data collection, cleaning, and formatting. Our goal is to provide a benchmark that is not only comprehensive and multi-temporal, but also easy to reproduce and extend for future studies. Figure 1 illustrates the complete construction pipeline.

**Dataset Construction.** Our dataset integrates three complementary sources to capture the textual, structural, and temporal dimensions of scientific publications. arXiv provides large-scale metadata

(titles, authors, affiliations, abstracts, publication dates), ar5iv supplies HTML-converted papers for structured extraction of full text and multimodal elements (e.g., figures, tables), and Semantic Scholar offers citation data with accurate temporal stamps. From these sources, we sample 1.5M papers across disciplines, extract references and citation subgraphs, and remove entries with incomplete or misindexed timestamps. To reduce noise in the metadata (e.g., venues, affiliations), we apply Qwen3-8B for filtering and normalization, and perform stratified sampling to balance disciplinary distributions (details in Appendix). The final release provides cleaned metadata, citation graphs, and modular tools for parsing, cleaning, and formatting, enabling reproducible construction of both large-scale and multi-temporal subsets.

**Dataset Split.** Citation-based prediction tasks are typically formulated in one of three ways: (i) forecasting citations at a single future time point (e.g., HINTS (Jiang et al., 2021)), (ii) splitting data chronologically to predict multi-step future trajectories (Xue et al., 2024) and (iii) focusing on only content and treating every paper as newborn ones (Zhao et al., 2024a). Following their choices, we

Table 1: Dataset splits and graph statistics. Affiliation$^*$ and Venue$^*$ fields are normalized using LLM-based cleaning.

|  | Train | Valid | Test_Prev. | Test_Fresh |
|---|---|---|---|---|
| # Papers | 210,519 | 100,796 | 133,432 | 199,641 |
|  | Cite | Author | Affiliation$^*$ | Venue$^*$ |
| # Nodes | – | 825,548 | 16,787 | 13,393 |
| # Edges | 16,751,419 | 2,232,276 | 3,618,802 | 190,597 |

index both 5-month and 5-year citation counts upon publication as prediction targets. We partition the dataset by publication year into training, validation, and test splits, and ensure no paper overlap in training and testing. To stress-test generalization, the test set is further divided into a previous test split (temporally close to training) and a fresh test split (strictly forward-looking). The resulting dataset contains 1,081,339 focal papers, with a total of 4,738,195 referenced papers, spanning ten disciplines. Table 1 provides a detailed statistics of our constructed dataset.

**Comparison with Existing Datasets.** Table 2 situates our benchmark in the landscape of scientific paper datasets. Existing resources each contribute valuable aspects: APS Bai et al. (2019) and DBLP Zhu & Ban (2018) provide structured networks but lack multimodality; PubMed Fu & Aliferis (2008) offers scale but limited label granularity; S2AG [1] achieves massive coverage but with sparse annotations; Aminer [2] and Bio-Sci Zhang et al. (2023) capture recent slices but often full content. Our dataset is the first to unify multi-grained impact labels, multi-domain coverage, multimodal content, and built-in processing tools within a single resource. With over one million papers, it strikes a balance between scale and usability, enabling both large-scale training and varied temporal analysis.

Table 2: Comparison of popular impact prediction paper datasets and databases. Abbreviations: Gran. = multi-grained labels, Dom. = multi-domain coverage.

| Dataset | Year | Gran. | Dom. | Tools | #Papers |
|---|---|---|---|---|---|
| APS | – | ✗ | ✗ | ✗ | 0.45M |
| Pubmed | 2003 | ✗ | ✗ | ✗ | 1.10M |
| DBLP | 2008 | ✗ | ✗ | ✗ | 1.80M |
| S2AG | 2022 | ✓ | ✓ | ✗ | 205M |
| Aminer | 2023 | ✗ | ✗ | ✗ | 5.25M |
| Bio-Sci | 2023 | ✗ | ✓ | ✗ | 0.03M |
| **Ours** | 2025 | ✓ | ✓ | ✓ | 1.08M |

# 4 METHODOLOGY

In general, citation impact prediction has been approached from two directions. Graph-based methods (Jiang et al., 2021; Xue et al., 2024) (Figure 2(a)) encode papers and metadata as a heterogeneous graph, learning structural and textual features via GNNs before regression, but struggle with semantics and temporal dynamics. LLM-based methods (Zhao et al., 2024a) (Figure 2(b)) leverage large language models with lightweight tuning (e.g., LoRA) to capture rich text semantics, yet overlook graph and time signals. Our *LLM4Impact predictor* framework (Figure 2(c)) unifies both by injecting graph and temporal embeddings as prefix tokens into LLM prompts, enabling multi-feature fusion for accurate citation forecasting. Below, we elaborate on our framework details.

---

[1] https://www.semanticscholar.org/

[2] https://www.kaggle.com/datasets/kmader/aminer-academic-citation-dataset

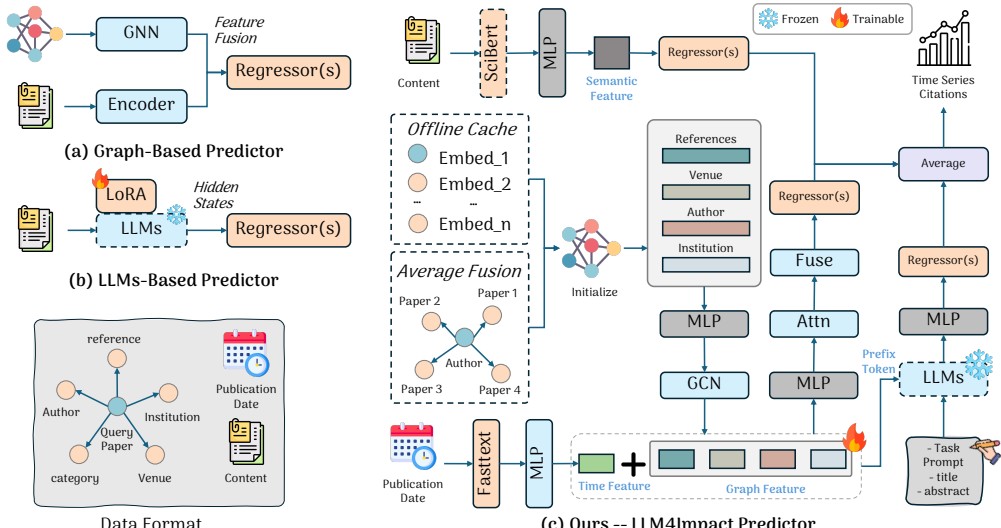

Figure 2: Overview of main scientific impact prediction framework: (a) graph baseline using a GNN with a separate text encoder for graph and semantic features; (b) LLM-only baseline focusing on semantic representations; (c) our proposed model that jointly and equally models heterogeneous graph structure, semantic text, and temporal signals with LLMs features.

## 4.1 TASK FORMULATION

We formalize the problem of scientific impact prediction as a heterogeneous graph learning task with temporal forecasting. Let the heterogeneous graph be denoted as $G = \langle N, E \rangle$ where $N$ is the set of nodes and $E$ is the set of edges. The node set $N$ consists of seven types of entities: *papers, references, authors, institutions, venues, category, and publication dates*. The edge set $E$ captures five types of relations, including $has\_author$, $has\_affiliation$, $published\_in$(venue), $cites$, $published\_at$(time). We treat all papers as newborn, i.e., the citation graph excludes existing citation counts for each query paper. Given a target paper $p \in N$, our objective is to predict its future citation across multiple temporal horizons. Specifically, the prediction target is defined as

$$Y_p = \{y_p^1, y_p^2, \ldots, y_p^L\},$$

where $y_p^l$ denotes the predicted citation count of paper $p$ under temporal scale $l$. In this work, we set $L = 5$, corresponding to both *yearly* and *monthly* citation scales over the future time scale. Formally, the task can be expressed as learning a mapping function

$$f : (G, p) \mapsto Y_p,$$

which leverages heterogeneous graph structure, semantic text, and temporal signals to estimate the long-term scientific impact of the paper.

## 4.2 INITIALIZATION

We begin by constructing semantic and temporal representations for each academic paper $P_i$. For semantics, we leverage SciBERT $\Phi_{SciBERT}(\cdot)$ to encode the title $T_i$ and abstract $A_i$, yielding an initial semantic embedding:

$$\mathbf{e}_{sem}^{(i)} = \Phi_{SciBERT}(T_i, A_i).$$

This embedding is jointly used in two roles: (i) as the input to the paper semantic predictor, and (ii) as the initialization of paper nodes in the citation graph. For other node types (authors, venues, and institutions), embeddings are initialized by aggregating the semantic embeddings of their neighboring paper nodes, ensuring that every node representation is grounded in paper semantics.

To incorporate temporal signals, we encode the publication timestamp $Y_i$ of $P_i$ using pretrained FastText model (Bojanowski et al., 2017) $\Phi_{FT}(\cdot)$, obtaining a dense temporal embedding:

$$\mathbf{e}_{time}^{(i)} = \Phi_{FT}(Y_i).$$

This embedding captures periodic and continuous time semantics and serves as the initialization of the time feature for each paper.

### 4.3 GRAPH-BASED FEATURE EXTRACTION

The scientific ecosystem is inherently relational, with citations, co-authorships, and venues shaping the trajectory of research impact. We model this ecosystem as the heterogeneous graph $G$ and for each feature type, a feature projector $\mathcal{P}_{feat}(\cdot)$ maps raw node embeddings into a common latent space. Subsequently, four-layers of Graph Convolutional Network (GCN) encoders $\mathcal{G}_{enc}(\cdot)$ refine these projected embeddings by propagating structural signals:

$$\mathbf{h}_{feat}^{(i)} = \mathcal{G}_{enc}\big(\mathcal{P}_{feat}(\mathbf{e}_{feat}^{(i)})\big).$$

where $\mathbf{e}_{feat}^{(i)}$ denotes the paper initial embeddings. The processed heterogeneous features across different types of nodes are then organized into a feature sequence $\mathbf{H}^{(i)} = \{\mathbf{h}_{feat}^{(i)}\}$ and concatenated with the projected temporal embedding $\mathbf{h}_{time}^{(i)}$. A cross-feature attention mechanism $\mathcal{A}(\cdot)$ models inter-feature dependencies, producing attended representations $\mathbf{H}'^{(i)}$. This design allows the model to dynamically highlight the most informative feature types for each paper, while down-weighting less relevant signals. Then, we apply global average pooling followed by a feature fusion network $\mathcal{F}(\cdot)$ to yield the final graph features:

$$\mathbf{e}_{graph}^{(i)} = \mathcal{F}\Big(\mathrm{Pool}\big(\mathcal{A}(\mathbf{H}^{(i)})\big)\Big).$$

Finally, we feed the graph features $\mathbf{e}_{graph}^{(i)}$ into five MLP layers, each dedicated to regressing the citation count within a specific future time scale $L$. To ensure numerical stability and constrain the prediction range, we apply a logarithmic transformation to the citation numbers before regression.

### 4.4 LLMs INTEGRATION

While graph embeddings capture relational and structural influence, understanding the future potential of a paper requires reasoning over semantic content and latent topics. To this end, we integrate LLMs with the graph encoder. Specifically, we employ a prefix-tuning strategy to inject graph-derived signals into the LLM. The graph embedding $\mathbf{e}_{graph}^{(i)}$ is projected into the LLM hidden space:

$$\mathbf{p}_{graph}^{(i)} = \mathcal{P}_{graph}(\mathbf{e}_{graph}^{(i)}),$$

and transformed into a sequence of prefix tokens $\mathbf{P}^{(i)} \in \mathbb{R}^{L_p \times d_{LLM}}$, where $L_p$ is the prefix length. These prefix tokens are concatenated with the LLM's input sequence embeddings (title and abstract, and also task prompt), effectively conditioning the LLM on structural knowledge.

The combined input is processed through the LLM's transformer layers, producing a mixed representation $\mathbf{h}_{LLM}^{(i)}$ that encodes both semantic understanding and structural context:

$$\mathbf{h}_{LLM}^{(i)} = \Phi_{LLM}\big([\mathbf{P}_{graph}^{(i)}\|(T_i, A_i)]\big).$$

This design balances efficiency and effectiveness: prefix-tuning avoids fine-tuning the entire LLM while enabling task-specific adaptation. The final hidden representation $\mathbf{h}_{LLM}^{(i)}$ is then passed through lightweight adapters and MLP heads for impact prediction.

### 4.5 TRAINING

During training, we obtain predictions from the three feature types and then average their outputs to produce the final citation sequence prediction. Given $\mathbf{y}^{(i)}$ as ground-truth citation sequence of paper $P_i$, our training loss is to minimize all sequence differences:

$$\mathcal{L}_{MSE} = \frac{1}{N} \sum_{i=1}^{N} \left\| \hat{\mathbf{y}}^{(i)} - \mathbf{y}^{(i)} \right\|_2^2,$$

Table 3: Comparison of citation prediction models across yearly windows. * denotes LLMs are finetuned or evaluated on 10% ratio of dataset while keep reporting stable results. Please refer to Appendix E for scaling laws on dataset size. ft denotes LoRA fine-tuning.

| Model | Previous (RMSLE ↓) | | | | | | Fresh (RMSLE ↓) | | | | | |
|---|---|---|---|---|---|---|---|---|---|---|---|---|
| | Year_1 | Year_2 | Year_3 | Year_4 | Year_5 | Avg. | Year_1 | Year_2 | Year_3 | Year_4 | Year_5 | Avg. |
| *Traditional* | | | | | | | | | | | | |
| SciBert | 0.8278 | 0.9615 | 1.0499 | 1.1152 | 1.1639 | 1.0306 | 0.9614 | 1.0944 | 1.1743 | 1.2270 | 1.2541 | 1.1471 |
| Qwen-Embed. | 0.8830 | 1.0459 | 1.1497 | 1.2240 | 1.2790 | 1.1251 | 1.0407 | 1.2190 | 1.3110 | 1.3762 | 1.3960 | 1.2752 |
| HINTS | 1.0811 | 1.1602 | 1.2232 | 1.3180 | 1.3927 | 1.2534 | 1.0301 | 1.1844 | 1.2529 | 1.3483 | 1.4266 | 1.2485 |
| NAIP | 0.9894 | 0.9888 | 1.4546 | 1.8141 | 1.5828 | 1.3932 | 0.8652 | 0.9038 | 1.3458 | 1.6541 | 1.5495 | 1.2918 |
| *LLMs-based* | | | | | | | | | | | | |
| *Llama-3B | 0.8833 | 1.0170 | 1.1311 | 1.2064 | 1.2680 | 1.1012 | 1.0563 | 1.2884 | 1.3775 | 1.4268 | 1.5032 | 1.3304 |
| *Qwen-4B | 0.9597 | 1.1893 | 1.3701 | 1.5098 | 1.6197 | 1.3297 | 1.1650 | 1.3052 | 1.3809 | 1.4369 | 1.4718 | 1.3519 |
| *Qwen-4B(ft) | 0.9616 | 1.2371 | 1.4535 | 1.6127 | 1.7463 | 1.4022 | 1.1899 | 1.5586 | 1.7601 | 1.8911 | 1.9862 | 1.6772 |
| *Qwen-8B | 1.5001 | 1.8924 | 2.1054 | 2.2623 | 2.4476 | 2.0416 | 1.5458 | 2.0021 | 2.2427 | 2.3836 | 2.5070 | 2.1362 |
| **Ours** | **0.6772** | **0.6932** | **0.7809** | **0.8022** | **0.9177** | **0.7719** | **0.7170** | **0.7426** | **0.7892** | **0.8302** | **0.9514** | **0.7745** |

Table 4: Comparison of monthly citation prediction results. LLMs often overlook the monthly setting specified in the prompt, leading to consistently worse performance compared to other baselines.

| Model | Previous (RMSLE ↓) | | | | | | Fresh (RMSLE ↓) | | | | | |
|---|---|---|---|---|---|---|---|---|---|---|---|---|
| | Mon._1 | Mon._2 | Mon._3 | Mon._4 | Mon._5 | Avg. | Mon._1 | Mon._2 | Mon._3 | Mon._4 | Mon._5 | Avg. |
| *Traditional* | | | | | | | | | | | | |
| SciBert | 0.2091 | 0.2793 | 0.3369 | 0.3879 | 0.4332 | 0.3386 | 0.2266 | 0.3141 | 0.3885 | 0.4536 | 0.5135 | 0.3925 |
| Qwen3-Embed. | 0.2107 | 0.2811 | 0.3403 | 0.3939 | 0.4404 | 0.3430 | 0.2283 | 0.3145 | 0.3905 | 0.4623 | 0.5232 | 0.3978 |
| HINTS | 0.5111 | 0.6448 | 0.6842 | 0.7204 | 0.7311 | 0.6562 | 0.5452 | 0.5661 | 0.6154 | 0.6577 | 0.7200 | 0.6209 |
| NAIP | 0.4026 | 0.4773 | 0.9666 | 1.3654 | 0.9094 | 0.7915 | 0.3032 | 0.4036 | 0.8110 | 1.1097 | 1.0898 | 0.7098 |
| *LLMs-based* | | | | | | | | | | | | |
| *Llama-3B | 0.6568 | 1.0175 | 1.2422 | 1.4531 | 1.6029 | 1.1945 | 0.6669 | 0.9927 | 1.2825 | 1.4835 | 1.5942 | 1.2040 |
| *Qwen-4B | 0.8741 | 0.9800 | 1.0861 | 1.1707 | 1.2221 | 1.0666 | 1.1650 | 1.3052 | 1.3809 | 1.4369 | 1.4718 | 1.3519 |
| *Qwen-4B (ft) | 0.2334 | 0.2913 | 0.3690 | 0.4217 | 0.4744 | 0.3579 | 1.4683 | 1.8955 | 2.1140 | 2.2625 | 2.3635 | 2.0207 |
| *Qwen-8B | 0.3470 | 0.4880 | 0.5705 | 0.5947 | 0.6128 | 0.5226 | 0.1478 | 0.3470 | 0.4186 | 0.6512 | 0.7898 | 0.4709 |
| **Ours** | **0.1845** | **0.1861** | **0.2128** | **0.2226** | **0.4539** | **0.2520** | **0.2117** | **0.2468** | **0.2396** | **0.2831** | **0.4673** | **0.2897** |

## 5 EXPERIMENT AND ANALYSIS

### 5.1 EXPERIMENT SETTING

**Baselines** We compare our approach against four representative baselines. SciBERT is a pre-trained model specialized for scientific text, capturing content-based features. Qwen-Embed. is the newest encoder model that captures more accurate semantic representations (Zhang et al., 2025). HINTS (Jiang et al., 2021) models dynamic heterogeneous information networks for citation time series prediction. NAIP (Zhao et al., 2024a) is a state-of-the-art method that integrates LLMs for the prediction of direct impact. To further explore LLM usage, we additionally prompt and fine-tune several LLMs on this task, including Llama3.2-3B (Dubey et al., 2024), Qwen-4B, and Qwen-8B (Yang et al., 2025). Due to the large scale of our dataset, we exclude models with more than 8B.

**Evaluation Metrics** We evaluate model performance using two common metrics. RMSLE (Root Mean Squared Logarithmic Error) is a regression metric suitable for highly skewed count data such as citations, penalizing relative errors. NDCG@20 (Normalized Discounted Cumulative Gain at 20) is a ranking metric that assesses the quality of predicted top-20 citation lists, giving higher weight to correctly ranked items at the top. We provide NDCG@20 results in Appendix B.

### 5.2 MAIN RESULTS AND LLM INTEGRATION (RQ1)

Tables 3 and 4 detail the superior performance of our proposed framework across all yearly and monthly citation prediction windows for both previous and fresh test sets.

**Overall results.** Our method consistently outperforms both traditional baselines (e.g., SciBERT, HINTS, NAIP) and LLMs (LLaMA-3B, Qwen-4B/8B in zero-shot and finetuned settings) in citation prediction. On the Previous set, it achieves substantially lower average RMSLE, demonstrating that

task-specific design yields the most significant error reduction for yearly prediction. On the Fresh set, our approach maintains the best performance, highlighting stronger robustness to distribution shifts. Monthly-level prediction further exhibits lower errors than yearly-level prediction, demonstrating that finer temporal granularity is easier to model; yet our method consistently achieves the lowest RMSLE across both Previous and Fresh sets, confirming its advantage at fine-grained temporal scales. The degradation of baselines on the Fresh set reflects pronounced distribution shift, while our method shows the smallest increase in mean RMSLE, indicating superior out-of-distribution and temporal extrapolation robustness. We also note a natural trend that papers closer to the present tend to have higher average citations, inherently producing a distribution shift relative to the training set; a visualization of this distribution is provided in Appendix A.

**LLMs integration.** Across two test sets, LoRA finetuning is worse than our method in both temporal scales and sometimes even underperforms the corresponding zeroshot prompts, implying that lightweight finetuning struggles to encode temporal and cumulative-citation structure and may introduce overfitting or prompt misalignment. These observations align with recent evidence in sequence or time-series modeling Tan et al. (2024): general-purpose LLMs are better leveraged as text encoders for hidden-state capture, while temporal reasoning and prediction should be delegated to task-specialized architectures and feature engineering (e.g., graph structure and historical trajectory statistics), validating the design choice in our framework. In addition, we plot the prediction distribution of LLMs in zero-shot and LoRA finetuning setting in Figure 3(b). We can infer that for temporal task modeling, fine-tuning LLMs tends to overfit training-specific values and ignore growth patterns, leading to underestimation, whereas graph-based and historical trajectory methods are better suited for cumulative citation prediction.

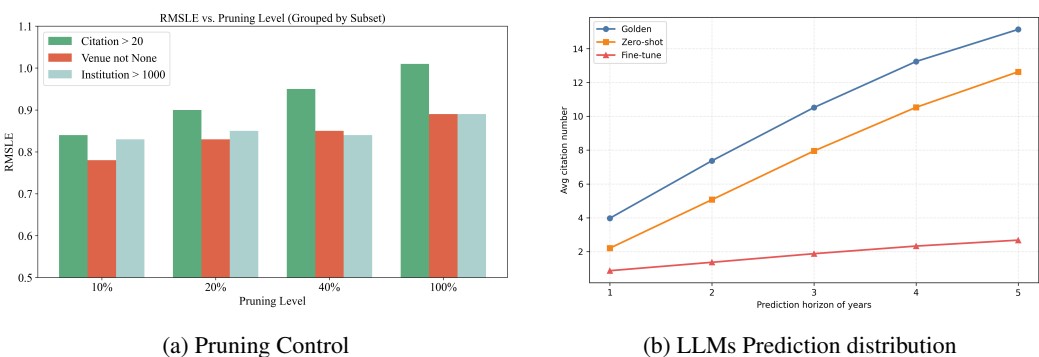

(a) Pruning Control          (b) LLMs Prediction distribution

Figure 3: Finetuning Comparison of LLMs after two thousand samples iteration, on year level.

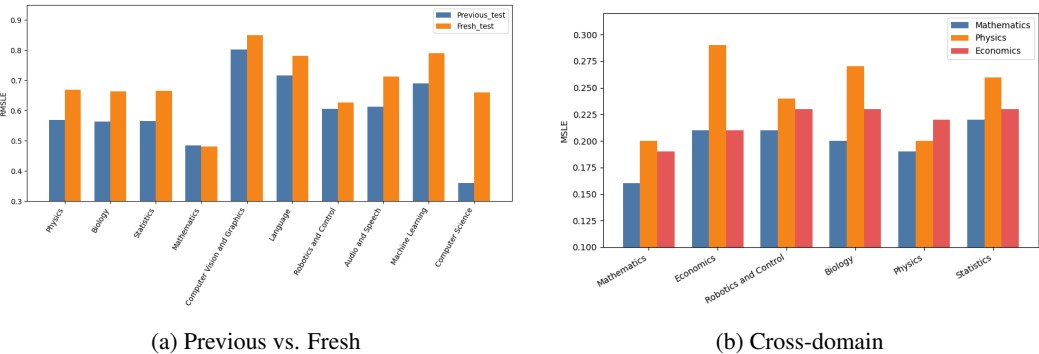

(a) Previous vs. Fresh          (b) Cross-domain

Figure 4: Control experiment and cross-domain experiment. (a) RMSLE comparison between previous and fresh test sets across scientific domains shows consistently higher errors on fresh data, especially in machine learning and language. (b) Cross-domain evaluation, where models are trained on one domain and tested on all, highlights substantial performance drops under domain shift.

## 5.3 SCIENTIFIC FACTORS (RQ2).

To assess the contribution of different graph attributes, we performed ablation experiments by progressively pruning three types of nodes: highly cited references (Citation > 20), venue information (Venue not None), and large institutions (Institution > 1000) at 10%, 20%, 40%, and 100% in the test set. The results (Figure 3(a)) show that model performance consistently degrades as pruning increases, with reference nodes having the largest impact, as full removal raises RMSLE above 1.0. Venue and institution nodes also contribute to performance, but to a lesser extent, indicating that direct reference links are the most critical features for accurate scientific impact prediction, while venue and institution provide complementary contextual information.

We further conducted cross-disciplinary knowledge transfer experiments as shown in Figure4(b), training models on a source discipline and evaluating them across multiple target disciplines. Results indicate that models trained on Mathematics are the most robust, achieving the lowest or near-lowest MSLE across most target disciplines; models trained on Physics generalize the worst, exhibiting the highest errors universally. Besides, the relative difficulty across target disciplines is consistent: Physics and Biology exhibit higher MSLE, suggesting noisier or more heterogeneous citation growth, while Statistics and Mathematics remain comparatively stable.

## 5.4 MORE ANLAYSIS AND DISCUSSION

Table 5: Comparison of RMSE scores between two impact measurement method along with their correlation scores *Cor.* (all are statistically significant). C_log means using log citation number and C_TNCSI means using TNCSI score.

| Metric | Year_1 | Year_2 | Year_3 | Year_4 | Year_5 | Avg |
|--------|--------|--------|--------|--------|--------|-----|
| *Previous* | | | | | | |
| C_log | 0.6772 | 0.6932 | 0.7809 | 0.8022 | 0.9177 | 0.7719 |
| C_TNCSI | 0.2016 | 0.2211 | 0.2291 | 0.2327 | 0.2352 | 0.2239 |
| Cor. | 0.7900 | 0.8061 | 0.8062 | 0.8119 | 0.8135 | 0.8055 |
| *Fresh* | | | | | | |
| C_log | 0.7170 | 0.7426 | 0.7892 | 0.8302 | 0.9514 | 0.7745 |
| C_TNCSI | 0.2316 | 0.2441 | 0.2556 | 0.2556 | 0.2532 | 0.2480 |
| Cor. | 0.8004 | 0.8128 | 0.8095 | 0.8136 | 0.8128 | 0.8098 |

**Other Metrics.** We further investigated the impact of alternative metrics on model evaluation. Following prior work, we extended our prediction target to Topic Normalized Citation Success Index(TNCSI) (Zhao et al., 2024c), which normalizes citation counts relative to the citation distribution of publications in the same field, thereby mitigating cross-disciplinary citation bias. Table 5 presents the rmse scores via using both raw citation counts and the TNCSI metric, and their spearman correlation coefficients (Spearman, 1961). We observe that despite differences in relative scores, the corresponding predictions exhibit a strong positive correlation. Moreover, we find that the fresh test set continues to pose greater challenges compared to the previous one.

**Domain Differences.** Figure 4(a) visualizes the error distribution across disciplines, revealing substantial performance gaps between fields. Moreover, performance does not exhibit a simple positive correlation with the number of training samples; for instance, Physics has the largest number of training samples (see Appendix A) but does not achieve the best performance, indicating that task difficulty and distribution shift primarily drive the observed differences.

## 6 CONCLUSION

This work tackles the challenging problem of forecasting scientific impact across multiple temporal horizons and heterogeneous data sources. We introduce a unified framework that integrates heterogeneous graph neural networks with LLMs, supported by a large-scale and multi-temporal benchmark dataset tailored for comprehensive and fine-grained evaluation. Through extensive empirical analysis, our approach consistently outperforms existing methods, achieving state-of-the-art performance across both yearly and monthly prediction horizons, evaluated on both Previous and Fresh test subsets. These findings highlight the importance of jointly modeling structural dependencies captured by heterogeneous graphs and rich semantic information encoded in textual content for reliable and generalizable impact prediction. Future work could explore deeper integration of multimodal data and adaptive model architectures to further incorporate more domain specific factors, such as reproducibility in computer science and scalable investigation in sociological research.

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

# Appendix

## Table of Contents

## A    DATASET CONSTRUCTION AND STATISTICS

### A.1    METADATA STATISTICS

The distribution of our dataset is illustrated in Figure 5. The left panel shows the sampling and partitioning across different disciplines. Following the proportions of the real-world distribution, we sample accordingly, resulting in uniformly distributed subsets without noticeable anomalies.

Figure 5b further depicts the number of publications from 1983 to 2019, which exhibits a rapid year-by-year increase. In particular, the growth becomes exponential after 2010, with the annual publication volume surpassing seventy thousand papers in 2019.

### A.2    SPLIT SET DISTRIBUTION

In addition, we visualize the distribution of the golden citation subset in Figure 6. Overall, the citation trajectories across different datasets are similar. However, the Fresh set demonstrates substantially higher average citation counts compared to other categories. This gap is especially pronounced in the yearly distribution, where the Fresh set grows more rapidly, while the monthly distribution exhibits relatively lower overall citation values.

## B    NDCG AND TNCSI CALCULATION

### B.1    NDCG ANALYSIS

We report the evaluation results using the NDCG@20 metric, as shown in Table 6 (Järvelin & Kekäläinen, 2002). Compared with the RMSLE metric, the conclusions drawn from NDCG@20 appear to be more stochastic. For instance, the metric does not necessarily show a consistent decline in performance as the interval length increases. This indicates that the loss function employed in our task is only effective for optimizing regression objectives, but not well-suited for ranking tasks.

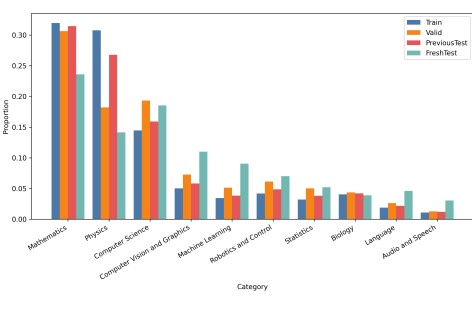

(a) Paper distributions among categories.

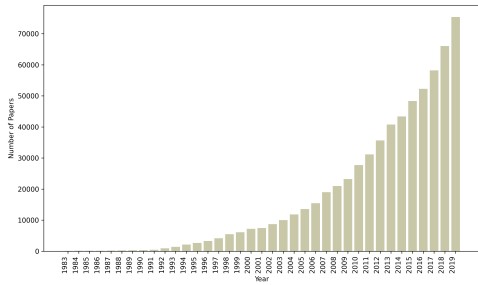

(b) Paper number distribution among years.

Figure 5: Dataset statistics: distributions of papers by research categories and publication years.

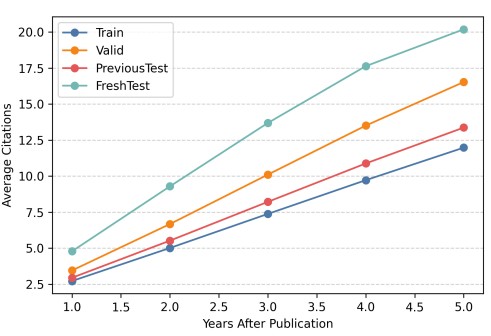

(a) Citation distribution among years.

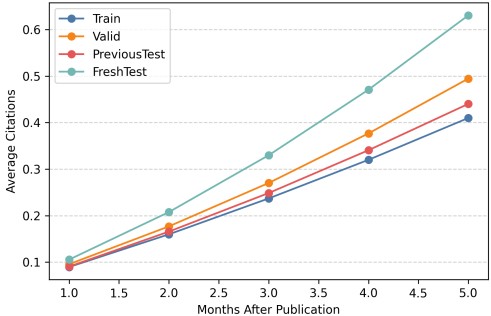

(b) Citation distribution among months

Figure 6: Distributional statistics of the three datasets. The results demonstrate that their citation dynamics are positively correlated and exhibit similar temporal trends, supporting the comparability of evaluation across datasets.**We show that the distributions across categories in our dataset are similar, enabling the model to learn and predict effectively; citation counts for fresh test papers are higher than for previous ones, while the average citation values at the month level are lower.**

### B.2 TNCSI

TNCSI (Topic Normalized Citation Success Index) is a metric designed to evaluate the citation impact of articles by comparing them within the same research field, providing a normalized score between 0 and 1 that indicates the likelihood an article's impact surpasses that of its peers. Unlike traditional citation counts, TNCSI normalizes across fields but originally focused on review papers and cumulative citations, limiting its suitability for comparing newly published or regular research articles. The fomulation is as follows:

Table 6: NDCG@5 scores of citation prediction models across yearly windows (Previous vs. Fresh).

| Model | Previous (NDCG@20) ↑ | | | | | | Fresh (NDCG@20) ↑ | | | | | |
|---|---|---|---|---|---|---|---|---|---|---|---|---|
| | Year_1 | Year_2 | Year_3 | Year_4 | Year_5 | Avg. | Year_1 | Year_2 | Year_3 | Year_4 | Year_5 | Avg. |
| SciBert | 0.3668 | 0.3255 | 0.4330 | 0.4609 | 0.5374 | 0.5374 | 0.2844 | 0.3965 | 0.3999 | 0.4075 | 0.2982 | 0.2958 |
| Qwen-Embed. | 0.1721 | 0.2141 | 0.2929 | 0.3111 | 0.2586 | 0.2586 | 0.1965 | 0.2374 | 0.2576 | 0.2683 | 0.2739 | 0.2717 |
| HINTS | 0.1431 | 0.2698 | 0.3287 | 0.3174 | 0.3847 | 0.2891 | 0.1290 | 0.1824 | 0.2159 | 0.1597 | 0.2705 | 0.1915 |
| NAIP | 0.3173 | 0.5067 | 0.3695 | 0.7059 | 0.4285 | 0.3475 | 0.3134 | 0.4217 | 0.5705 | 0.4554 | 0.4402 | 0.3636 |
| *Qwen-4B | 0.5933 | 0.7498 | 0.7789 | 0.7769 | 0.7840 | 0.7366 | – | – | – | – | – | – |
| *Qwen-4B(ft) | 0.8709 | 0.9905 | 0.9851 | 0.9806 | 0.9715 | 0.9597 | – | – | – | – | – | – |
| Ours | **0.6963** | **0.7021** | **0.6822** | **0.7132** | **0.7392** | **0.7066** | **0.6638** | **0.6367** | **0.6313** | **0.6159** | **0.6764** | **0.6393** |

$$TNCSI = \int_0^{\text{cites}} \lambda e^{-\lambda x}, dx, \quad x \geq 0. \tag{1}$$

## C  TRAINING PARAMETERS

We trained our model on one single H100 or A100 GPU. We configure our hybrid model with a GCN-based graph encoder (3 layers, hidden size 256, 8 heads, dropout 0.1), a Qwen-based LLM backbone (`Qwen/Qwen3-0.6B` by default, with adapters of dimension 64), and a learnable prefix embedding of length 10 and dimension 512. The prediction head is a multi-layer perceptron with hidden sizes [512, 256, 128], attention enabled, and regression as the default prediction type. Training is performed with a batch size of 16, learning rate of $2 \times 10^{-5}$, weight decay of 0.01, warmup ratio of 0.1, for up to 20 epochs with early stopping (patience 5, $\delta = 10^{-4}$). Input sequences are truncated at 512 tokens, with at most 10 authors and 5 institutions retained. For completeness, the full set of hyperparameters and configuration files will be available in our released code repository.

## D  THE USE OF LARGE LANGUAGE MODELS

Large language models are used in our early-stage writing for wording and grammar checking, as well as searching for missing literature. LLMs are not involved in the later iterations of the paper writing. Therefore, we do not consider LLMs as significant contributors to this paper.

## E  LLMs INTEGRATION AND DETAILS

In this project, LLMs primarily serve three roles. First, they are employed for denoising the data, for which we directly utilized Qwen-4B. Second, we fine-tuned LLMs to predict citation sequences at the text level; however, the performance was adversely affected by noise. Finally, we integrated graph features to fine-tune prefix tokens, which constitutes the core innovation of our work.

Below, we also provide our used prompt template, including LLMs task prompt when doing prefix-tuning, data cleaning when doing data collection.

### E.1  LLMs LoRA FINETUNING

Furthermore, we attempted direct fine-tuning of LLMs at the text level. Although the training process converges (as shown in Figure 6), the performance is inferior compared to zero-shot and other baselines. An analysis of the erroneous cases further reveals that, after fine-tuning, LLMs exhibit a tendency to predict smaller values, likely as a conservative strategy. Figure 7b demonstrates that our sampling strategy yields stable experimental results.

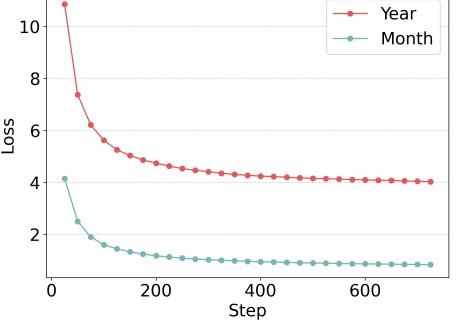
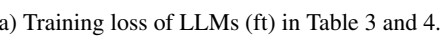

(a) Training loss of LLMs (ft) in Table 3 and 4.

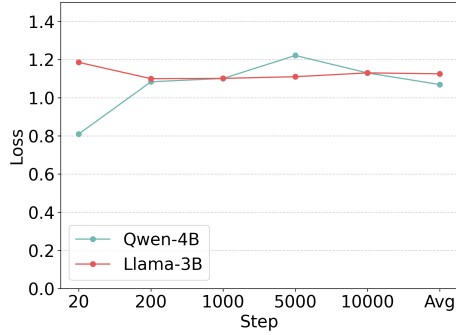

(b) Paper number distribution among years.

Figure 7: The loss curve of training LLMs and evaluation scaling laws along with sample numbers.

## E.2 PROMPT

---

**The prompt of LLMs to predict future citation counts.**

You are a scientometrics analyst. Task: Predict the future citation count of a paper within the specified horizon.

- Base your prediction **ONLY** on the provided metadata and abstract.
- Assume today's date is: {timestamps}.
- The prediction target is the total citations in future 1,2,3,4,5 years after publication (C5).
- Output a single line of five non-negative integers separated by commas.
- Return **ONLY the numbers**. No extra text or explanation.
- If information is missing, make a best estimate without adding new facts.

**Paper Metadata:**
Title: {title}
Authors: {author_list}
Institutions: {institutions}
Venue: {venue}
Field(s): {fields}
Abstract: {abstract}

**Example Output:**
1, 2, 3, 4, 5
**Generation Output:**
Please predict the five citation numbers accordingly.

---

**The prompt of LLMs to normalize noisy affiliations.**

You are given a list of affiliation strings. Some are valid institutions, some are duplicated, and some are meaningless. Please normalize them as follows:

- If the string is a valid institution name but with formatting issues, fix it (e.g., remove extra punctuation, unify into "Institution Name").
- If the string is a duplicate of another name, map it to the same corrected name.
- If the string is not an institution name (number, meaningless), map it to "unknown".
- Return the result strictly as a JSON dictionary, where each original string is mapped to its corrected normalized name.

**Example Input:**
["University of California, Los Angeles", "University of California, Los Angeles", "01 Collaboration"]

**Example Output:**

```
{
  "University of California, Los Angeles": "University of
      California",
  "01 Collaboration": "unknown"
}
```

**Generation Output:**
Please normalize the following affiliation list accordingly.

**The prompt of LLMs to normalize noisy venues.**

You are given a list of venue strings. Some are valid venues, some are duplicated, and some are meaningless. Please normalize them as follows:

- If the string is a valid venue name but with formatting issues, fix it (e.g., remove extra punctuation, use proper title case like "Venue Name").
- Remove any explicit year tokens (e.g., "2015", "2014").
- Remove edition indicators such as ordinal numbers or words (e.g., "1st", "2nd", "Third", "Fourth").
- If the venue contains a trailing parenthetical acronym/abbreviation like "(ICAC3N)", "(HPEC)", "(IVAPP)" that is all caps/digits/hyphen and ≤10 chars, drop that parenthetical part. Keep descriptive tracks in parentheses (e.g., "(Emerging Technologies)") as is.
- If both a full spelled-out name and an acronym appear together, keep only the full spelled-out name and drop the acronym.
- If the string is a duplicate of another (after normalization), map it to the same corrected name.
- If the string is not a venue name (pure number or meaningless), map it to "unknown".
- Return the result strictly as a JSON dictionary, where each original string is mapped to its corrected normalized name. Do not include comments or extra text.

**Example Input:**
["Journal of Machine Learning Research", "2015 IEEE International Conference on Web Services", "4th International Conference on X (ICACX)", "01 Collaboration"]

**Example Output:**

```
{
    "Journal of Machine Learning Research": "Journal of Machine
        Learning Research",
    "2015 IEEE International Conference on Web Services": "IEEE
        International Conference on Web Services",
    "4th International Conference on X (ICACX)": "International
        Conference on X",
    "01 Collaboration": "unknown"
}
```

**Generation Output:**
Please normalize the following venue list accordingly.

