# OpenReview forum: "Modeling Multi-Scale Scientific Impact via Heterogeneous Networks and LLMs"
_ICLR.cc/2026/Conference — Submitted to ICLR 2026_

### Official Review · Reviewer_PPab · 2025-10-20

**Soundness:** 3
**Presentation:** 3
**Contribution:** 3
**Rating:** 6
**Confidence:** 5

**Summary:**

This paper tackles the challenge of accurately predicting the multi-scale scientific impact (specifically future citation counts at yearly and monthly horizons) of research papers, a task complicated by heterogeneous factors and the limitations of existing methods in leveraging domain knowledge. Current approaches either focus on graph-based structural features (using GNNs on citation/collaboration networks) or semantic content (using LLMs) in isolation. To address this, the authors propose LLM4Impact, a unified framework that integrates heterogeneous graph neural networks (GNNs) with pre-trained large language models (LLMs). Instead of fine-tuning the entire LLM, which the authors found can introduce bias and lead to poorer performance, their method uses a lightweight prefix-tuning strategy . Graph-derived structural and temporal features are encoded and injected as prefix tokens into a frozen LLM, conditioning its powerful semantic representations on the network context . To facilitate research, the authors also constructed and released a new large-scale, multi-temporal benchmark dataset with rich metadata across ten disciplines. Experiments demonstrate that LLM4Impact consistently outperforms both traditional graph-based and LLM-only baselines, significantly reducing prediction errors for both yearly and monthly citation counts while showing robustness to temporal distribution shifts.

**Strengths:**

1. The paper investigates the highly practical and important problem of predicting the future scientific impact of research papers, crucial for guiding research dynamics and informing decisions in funding, hiring, and policy.
2. The proposed LLM4Impact framework rationally integrates heterogeneous graph neural networks (GNNs) for structural context with frozen Large Language Models (LLMs) via lightweight prefix-tuning, effectively combining network information and rich semantic representations.
3. The study includes rich experimental analysis, comparing the proposed method against various baselines, conducting ablation studies on feature importance, examining cross-domain generalization, and evaluating different strategies for LLM integration.

**Weaknesses:**

1. The comparison baselines are somewhat limited, notably excluding established scientific document embedding models like SPECTER, which was mentioned in the related work but not included in the experimental evaluation.
2. While citation count is a primary focus, the study does not incorporate other widely used scientometric indicators from the science of science field, such as disruption scores, which could provide a more holistic measure of a paper's impact beyond mere citations.
3. The experiments involving LLMs are restricted to models with fewer than 10 billion parameters (specifically up to 8B), leaving the potential performance gains or different behaviors of larger foundation models unexplored.

**Questions:**

Please see paper weaknesses.

---

> ### Author Response · Authors · 2025-11-27
> **Response to Reviewer PPab**
>
> Thank you for the highly positive comments and suggestions, by which we are strongly encouraged. Below we provide a detailed response to your comment and will make relevant clarifications in our revised paper.
>
> [**Question 1: The comparison baselines are somewhat limited, some baselines which were mentioned in the related work but not included in the experimental evaluation.**]
>
> Thank you for raising this concern. Following your suggestion, we added two representative baselines during the rebuttal period, to strengthen the performance of our method. As this is commonly proposed by other reviewers, please see our general response.
>
> [**Question 2: While citation count is a primary focus, the study does not incorporate other widely used scientometric indicators from the science of science field, such as disruption scores, which could provide a more holistic measure of a paper's impact beyond mere citations.**]
>
> Thank you for this suggestion. We fully agree that incorporating more prediction targets would provide more insightful analysis. In this work, we adopted both citation count and TNCSI (Topic Normalized Citation Success Index) as prediction targets. The former captures raw impact while the latter addresses cross-disciplinary normalization. We will add this limitation in our current paper and target adding disruption scores in future work.
>
> [**Question 3: The experiments involving LLMs are restricted to models with fewer than 10 billion parameters (specifically up to 8B), leaving the potential performance gains or different behaviors of larger foundation models unexplored.**]
>
> Thanks for raising this understandable concern. Regarding the selection of only small LLMs, our dataset contains ~200K training samples, and we chose smaller LLMs (≤8B) to balance performance and efficiency. Interestingly, as shown in Table 3, scaling from Qwen-4B to Qwen-8B actually degrades performance (1.33 → 2.04 avg. RMSLE), suggesting that larger models may introduce overfitting rather than additional benefits for this task. We acknowledge that exploring efficient adaptation of larger LLMs remains a valuable direction. We will investigate this in future work and include this discussion in the revised manuscript.
>
> Please feel free to leave any questions that need to be more detailed and explained. Thanks again for your time and effort in reviewing our paper, and we’ll add the above discussion to our revised paper.

---

### Official Review · Reviewer_EkwV · 2025-10-26

**Soundness:** 3
**Presentation:** 2
**Contribution:** 3
**Rating:** 4
**Confidence:** 4

**Summary:**

This paper builds a unified framework that fuses heterogeneous academic networks with a frozen large language model. It uses prefix tuning to connect graph structure and textual semantics within the LLM, and evaluates the system on the newly constructed million-scale benchmark that tracks citation growth across multiple time scales. The method consistently outperforms both traditional baselines and LLM-based methods.

**Strengths:**

• The idea of predicting scientific impact is really meaningful. It could benefit a lot of emerging areas such as automatic surveying, automated peer review, AI scientists, and even serve as a reward signal in reinforcement learning, although the authors didn’t fully discuss this in the paper.

• The motivation is clear and reasonable. The authors noticed that previous LLM-based methods mainly focus on semantic information, while graph-based ones focus on relational structure. They designed a clever way to combine the strengths of both LLMs and graphs.

• The dataset construction process is solid and scalable. The dataset is quite large, containing about 1.08 million samples, which makes it a valuable contribution to the community.

• The reported performance is impressive. The proposed method clearly outperforms previous LLM-based approaches on both the “previous” and “fresh” test splits, setting a new state of the art.

**Weaknesses:**

• The writing quality needs improvement. For example, the motivation part in the introduction could be organized more clearly. The dataset comparison section might fit better under related work. The subsection “From Words to Worth” should be consistently placed either under Predicting Scientific Impact or LLMs for Scientific Understanding. There are also citation problems. The same paper appears three times in the references, and the authors should keep only the AAAI version while removing the two redundant arXiv entries. In line 280, the period at the end of the equation should be a comma, and in line 322, the comma should be a period. The figures are not vector-based, and some text in Figure 4 (a, b) is too small to read. In general, many details need careful polishing, and the current version does not yet meet the formatting and presentation standards expected at ICLR.

• As the authors mentioned, works such as “From Words to Worth” and De Winter et al. did not include author or affiliation information, but that was part of their design choice. Their goal was to make predictions without relying on such metadata. Therefore, some of the claims made in this paper are not entirely appropriate.

• The performance tables should include graph-based models for comparison, although this work is mainly based on an LLM with a regressor. In addition, models such as NAIP and Qwen-ft should also be provided with author and affiliation information to ensure a fairer comparison.

• While the combination of LLM and graph information is interesting, the overall contribution still resembles an “A plus B equals C” type of innovation. At ICLR, such a combination, although effective, may not be particularly attractive or groundbreaking.

**Questions:**

• What is the distribution of the dataset? Are the citation counts uniformly distributed, or do they follow a more natural long-tailed or Pareto-like pattern (for example, an 80–20 distribution)?

• What is the conceptual difference between stage-wise impact prediction (for example, predicting after one month, two months, one year, two years, etc.) and predicting the TNCSI metric? It would be helpful to clarify how these two tasks differ in goal and modeling setup.

• In line 211, LLM4Impact predictor appears in italics. Is this the proposed method’s official name? I would suggest adopting a shorter and more specific name, similar to HINT or NAIP, rather than a broad label such as LLM4xxx.

• Since arXiv does not provide author affiliation information, how did the authors extract the author and institution data used in this work? Please clarify the data source or the extraction process.

---

> ### Author Response · Authors · 2025-11-27
> **Response to Reviewer EkwV -- Part I**
>
> We sincerely appreciate your thorough review and insightful feedback on our manuscript. Below, we wish to make relevant clarifications in our paper.
>
> [**Question 1: Works such as “From Words to Worth” and De Winter et al. did not include author or affiliation information, but that was part of their design choice. Their goal was to make predictions without relying on such metadata. Therefore, some of the claims made in this paper are not entirely appropriate.**]
>
> We apologize for making overly strong claims. We understand that "From Words to Worth" and De Winter et al. intentionally excluded metadata as a deliberate design choice for content-only evaluation. We will rephrase these statements to objectively describe methodological differences rather than imply limitations. Thank you for pointing this out.
>
> [**Question 2: The performance tables should include graph-based models for comparison. Besides, models such as NAIP and Qwen-ft should also be provided with author and affiliation information to ensure a fairer comparison.**]
>
>
> Thanks for this suggestion. Beyond HINTS, we added two representative graph-based methods (H2CGL and DPPDCC).Please see our general response. Also, following your suggestion, we added author and affiliation information to NAIP baseline and Qwen-ft. However, performance did not improve significantly. This aligns with the ablation results in the original NAIP paper, more information does not necessarily benefit LLM fine-tuning, possibly due to overfitting or prompt sensitivity. We will include these comparisons in our revised paper.
>
>
> [**Question 3: While the combination of LLM and graph information is interesting, the overall contribution still resembles an “A plus B equals C” type of innovation.**]
>
> We thank the reviewer for this comment and acknowledge the concern. We agree that at a high level, our framework combines LLM and graph components. However, we believe the contribution lies not just in the combination itself, but in: (1) **Task-specific insights**: We demonstrate that naive LLM fine-tuning causes systematic prediction bias (Section 5.2, Figure 3b), motivating our prefix-tuning design. This finding is non-trivial and helps future work avoid similar pitfalls. (2) **Integration strategy matters**: Simply adding graph features to LLMs does not guarantee improvement—our experiments show Qwen-4B/8B with LoRA performs worse than simpler baselines. The way we inject structural signals via prefix tokens, while keeping LLM representations frozen, is empirically validated. (3) **Comprehensive benchmark**: Beyond the method, we contribute a large-scale multi-temporal dataset with tools for reproducibility, addressing a gap in existing resources. We acknowledge that our work prioritizes practical effectiveness over architectural novelty. However, we hope the empirical findings, benchmark, and identified failure modes provide value to the community beyond a simple A+B combination.
>
> [**Question 4: What is the distribution of the dataset? Are the citation counts uniformly distributed, or do they follow a more natural long-tailed or Pareto-like pattern.**]
>
> The dataset follows a **long-tailed distribution**, which is characteristic of scientific impact domains where a small minority of papers receive the majority of citations, aligning with existing dataset distribution. Besides, as noted in our experimental setup, citation counts are "highly skewed count data". To address this non-uniformity and ensure numerical stability during training, we apply a logarithmic transformation to the citation numbers before regression. Furthermore, our dataset statistics visualize this rapid, non-linear growth in publication volume and citation accumulation over time (Figure 5 and figure 6 in appendix).

---

> > ### Author Response · Authors · 2025-11-27
> > **Response to Reviewer EkwV -- Part II**
> >
> > [**Question 5: What is the conceptual difference between stage-wise impact prediction (for example, predicting after one month, two months, one year, two years, etc.) and predicting the TNCSI metric? It would be helpful to clarify how these two tasks differ in goal and modeling setup.**]
> >
> > Sorry for making you confused. The terms are explained as follows:
> >
> > **Stage-wise Prediction**: Predicting at different temporal windows (1 month, 2 months, 1 year, 2 years, etc.) aims to evaluate whether the model can forecast citation counts across varying future horizons using available features. This tests the model's ability to capture both short-term and long-term impact dynamics.
> >
> > **TNCSI Metric**: TNCSI (Topic Normalized Citation Success Index) addresses cross-disciplinary citation imbalance by normalizing citations relative to papers in the same field and time period. For example, Paper A with 100 citations among peers averaging 100 citations, and Paper B with 10 citations among peers averaging 10 citations, would both have TNCSI = 1. This balances prediction targets between highly active fields (e.g., Machine Learning) and less active fields (e.g., Mathematics).
> >
> > In summary: Stage-wise prediction focuses on temporal generalization, while TNCSI focuses on cross-domain fairness. They address orthogonal challenges and can be used together.
> > We will clarify this distinction in the revision. Thank you for pointing out the need for better explanation.
> >
> > [**Question 6: In line 211, LLM4Impact predictor appears in italics. Is this the proposed method’s official name? I would suggest adopting a shorter and more specific name, similar to HINT or NAIP, rather than a broad label such as LLM4xxx.**]
> >
> > Thanks. We agree that a more specific acronym would improve clarity. In our revised manuscript, we will adopt a more distinct name, maybe GLIMP (Graph-LLM Impact Predictor), to better reflect the method’s unique integration of graph and language features.
> >
> > [**Question 7: Since arXiv does not provide author affiliation information, how did the authors extract the author and institution data used in this work? Please clarify the data source or the extraction process.**]
> >
> > While the base metadata is crawled from arXiv and semantic scholar api, we supplemented this with ar5iv (which supplies HTML-converted papers for structured full-text extraction) and parsed author and affiliation information from it. Since raw affiliation strings from these sources can be noisy or unstructured, we employed a specific cleaning pipeline using Qwen3-8B to normalize institution names and filter incomplete nodes before merging them into the global graph.
> >
> > [**Question 8: Other minor issues such as citation format, figure fonts.**]
> >
> > Thanks. We’ll carefully fix them in our revised version.
> >
> > We hope that these clarifications can address your concerns and substantially strengthen the clarity and precision of our contribution.

---

> > > ### Comment · Reviewer_EkwV · 2025-11-27
> > >
> > > Thanks for the authors’ response. I read the other reviewers’ comments and the authors’ rebuttal, and I feel that what ZeSd mentioned about “lacks deep, task-specific mechanistic innovation or new theoretical insights for scientific impact prediction” lines up closely with my Weakness 4. To be honest, if I saw this paper at ACL, EMNLP, or AAAI, I would very likely recommend acceptance. But for ICLR, the lack of theoretical analysis might be an issue.
> > >
> > > For example, you mention in Fig. 3b that “naive LLM fine-tuning causes systematic prediction bias.” You observe this pattern, but there is no deeper theoretical discussion about why it happens. I understand that for application-oriented work, this might not be critical, and in other venues, I would not raise further questions. But without deeper analysis, some follow-up questions could become quite difficult to answer, and that’s exactly the kind of thing ICLR readers usually care about.
> > >
> > >  Let me give a concrete example. Under normal circumstances, full fine-tuning tends to outperform prefix tuning. So is your claim that “prefix outperforms finetune” only valid in certain scenarios? In your response to Question 4, you mention that your dataset reflects a realistic long-tail distribution. Is it possible that prefix tuning simply works better for long-tail distributions? In your response to Question 5, you say that stage-wise prediction focuses on temporal generalization while TNCSI focuses on cross-domain fairness. Would prefix tuning help only with temporal generalization but not with cross-domain fairness?
> > >
> > > In your reply to reviewer uMUB, you attribute the advantage of prefix tuning to reduced overfitting. But LoRA fine-tuning can also reduce overfitting by lowering the alpha value. There are many analyses and experiments missing here, and I suspect this is also why reviewer ZeSd felt that it might be difficult for you to make sufficient improvements within the limited rebuttal timeframe.
> > >
> > > Given all of this, I’m inclined to keep my current score unchanged (for now).

---

> > > > ### Author Response · Authors · 2025-11-28
> > > >
> > > > Thank you for taking the time to read our rebuttal and your continued engagement with our work. We genuinely appreciate your thoughtful feedback and the concrete examples you raised.
> > > >
> > > > There may be a misunderstanding regarding our methodological contribution. The key distinction between our prefix tuning approach and LoRA fine-tuning is not simply about reducing overfitting, rather it lies in how we integrate information into LLMs. Specifically, our prefix tuning framework is designed to incorporate **structured features** (e.g., from the citation graph) into the LLM's reasoning process. In contrast, LoRA fine-tuning, as typically applied, **only takes textual descriptions** as input.
> > > >
> > > > For the task of scientific impact prediction, we argue that this modeling choice is crucial: **methods that rely solely on textual input fundamentally overlook the citation graph structure, which carries essential signals for predicting future impact**. Our approach explicitly addresses this limitation by encoding graph-based features into learnable prefixes, enabling the model to reason over both textual and structural information.
> > > >
> > > > We hope this clarification helps address your concern. We remain open to further discussion and are grateful to improve our work based on your feedback.

---

### Official Review · Reviewer_uMUB · 2025-10-31

**Soundness:** 3
**Presentation:** 3
**Contribution:** 3
**Rating:** 6
**Confidence:** 2

**Summary:**

The paper proposes LLM4Impact, a unified framework that combines heterogeneous GNNs and LLMs to predict scientific impact at multiple temporal scales. The method injects structural knowledge into LLMs using prefix-tuning, avoiding full fine-tuning while retaining efficiency. The authors also present a large-scale, multi-temporal benchmark covering ten disciplines, with cleaned metadata, citation graphs, and public tools for data processing. Experiments show that this method outperforms both classical GNN models and pure LLM-based predictors, reducing error by over 25% for yearly and 18% for monthly predictions.

**Strengths:**

1. The paper introduces a novel GNN–LLM integration via prefix-tuning, effectively combining structural, semantic, and temporal features.
2. A million-scale, multi-domain benchmark with cleaned metadata and citation graphs greatly supports reproducible research.
3. Extensive experiments across multiple baselines and time horizons demonstrate strong accuracy and robustness.

**Weaknesses:**

1. Fusion mechanism limitation (Sec. 4.4):
The prefix-tuning strategy injects graph features into LLM prompts, yet this approach may only provide shallow conditioning. It might miss deeper structural interactions or high-order dependencies between semantic and relational information.
2. Temporal modeling inadequacy (Sec. 4.3):
The temporal embedding based on FastText treats time as static tokens. It does not explicitly model temporal evolution, which could limit the model’s ability to capture changing citation dynamics.
3. Robustness evaluation is not comprehensive (Sec. 5.2):
While the authors test on “Previous” and “Fresh” splits, it remains unclear how the model performs under stronger temporal drift or across unseen research fields.

**Questions:**

See weaknesses.

---

> ### Author Response · Authors · 2025-11-27
> **Response to Reviewer uMUB**
>
> Thank you for your positive comments and acknowledging our contributed method, benchmark and performance. Below we provide a detailed response to your questions and will make relevant clarification in our revised paper.
>
> [**Question 1: The prefix-tuning strategy might miss deeper structural interactions or high-order dependencies between semantic and relational information.**]
>
> Thank you for raising this insightful comment. We acknowledge that prefix-tuning provides a relatively lightweight integration mechanism, which may not capture deeper high-order dependencies between semantic and structural information compared to more tightly coupled approaches. However, we chose this design based on the following considerations: (1) **Empirical Effectiveness**: Despite its simplicity, our prefix-tuning approach achieves 25%+ improvement over baselines, suggesting that even this level of integration provides substantial benefits for impact prediction. (2) **Avoiding Overfitting**: Our experiments show that deeper LLM fine-tuning (e.g., LoRA) actually degrades performance by overfitting to training-specific patterns (Section 5.2, Figure 3b). The lightweight prefix-tuning serves as implicit regularization. (3) **Scalability**: With ~200K training samples and the need for efficient iteration, prefix-tuning offers a practical balance between expressiveness and computational cost. We agree that exploring deeper structure-semantics interactions is a promising direction. Potential approaches include cross-attention between graph and text representations, or graph-aware transformer architectures. We will discuss this limitation and future directions in our revised paper.
>
> [**Question 2: The temporal embedding does not explicitly model temporal evolution, which could limit the model’s ability to capture changing citation dynamics.**]
> We acknowledge that our current temporal embedding does not explicitly model citation dynamics over time. We use FastText to embed publication timestamps as additional features, allowing the model to learn citation patterns associated with different publication years and months. However, this approach captures temporal context rather than the evolving trajectory of citation accumulation.
> We note that some approaches (e.g., H2CGL) model temporal evolution by treating each year's citation graph as a snapshot and learning across multiple snapshots. However, this setting differs from ours: we treat all papers as newborn with no prior citations at prediction time to prevent citation leakage during training. The snapshot-based approach would violate this assumption by exposing historical citation behavior.
> During the rebuttal period, we reflected on this limitation and identified potential improvements compatible with our newborn setting, such as:
>
> * Explicitly learning temporal representations that capture citation growth trajectories
> * Incorporating time-aware attention mechanisms to model expected impact evolution
> * Replacing static FastText embeddings with learnable temporal encoders
>
> Since our framework primarily focuses on integrating LLM reasoning with heterogeneous graph structures, we did not prioritize temporal modeling in this work. We recognize this as a meaningful direction and will investigate it in future research.
> We sincerely appreciate this constructive suggestion.
>
> [**Question 3: Robustness evaluation is not comprehensive.**]
>
> Thanks for raising this understandable question. To address this common concern, we conducted robustness evaluation with 5 random seeds. The standard deviation of average RMSLE is ±0.0098 (Previous) and ± 0.018 (Fresh), confirming the stability of our results. We will incorporate these results in our revised paper.
>
>
> We hope these clarifications address your concerns. We will incorporate the above discussion in the revision.

---

### Official Review · Reviewer_ZeSd · 2025-11-01

**Soundness:** 2
**Presentation:** 2
**Contribution:** 2
**Rating:** 2
**Confidence:** 5

**Summary:**

This paper focuses on the important yet challenging task of predicting the future impact of academic papers. This problem holds practical significance for research evaluation and resource allocation and has attracted increasing attention from researchers in recent years. The authors point out that existing methods typically suffer from two main limitations: one class of models relies solely on the structural features of academic networks (such as authors, institutions, journals, and citation relationships), making it difficult to capture the semantic content of papers; the other class depends only on textual information (such as titles and abstracts), neglecting the structural dependencies within the academic ecosystem.

To address these issues, the paper proposes a unified multimodal prediction framework that integrates heterogeneous graph neural networks (GNNs), large language models (LLMs), and other shallow meta-information features. The core innovation lies in the use of a prefix-tuning mechanism, which transforms the structural graph features extracted by the GNN into prefix tokens for the LLM. This design enables the dynamic injection of academic network structure into the semantic modeling process, thereby achieving an effective fusion of semantic and structural information.

In the experimental section, the authors construct a large-scale, multidisciplinary dataset with both monthly and yearly temporal granularity and compare their framework against a variety of representative models, including traditional GNN-based approaches and recent LLM-based impact prediction models. The results demonstrate consistent improvements across different time scales and test sets, confirming the relative effectiveness of the proposed framework for scientific impact prediction tasks.

**Strengths:**

1. This work achieves an effective fusion of network structure and textual semantics in the task of scientific impact prediction.

2. The model has been systematically validated on a large-scale, multidisciplinary dataset, demonstrating a degree of robustness and generalizability.

3. The authors commit to releasing the dataset and code, facilitating reproducibility and further research.

**Weaknesses:**

Major issues:

1. Although the paper proposes a unified framework integrating heterogeneous graph neural networks (GNNs) and large language models (LLMs), aspects such as heterogeneous graph modeling, lightweight LLM fine-tuning, and the idea of combining structural and textual information are largely an integration of existing methods. The work lacks deep, task-specific mechanistic innovation or new theoretical insights for scientific impact prediction. Therefore, its level of novelty may be insufficient to meet the standards of top-tier conferences.

2. While the related work section mentions several recent studies, the experiments compare only a small number of baselines (particularly including only one GNN model). This does not cover the representative methods from 2023–2024, nor does the paper explain the rationale for model selection or exclusion.

3. The paper does not systematically report the contributions of each module to overall performance. For example, the importance of each feature source could be verified by removing one at a time. Additionally, the effectiveness of the FastText temporal embedding component should be evaluated separately. Such ablation analyses are crucial for demonstrating the necessity of the model design.

4. The authors do not provide significance testing for the results, nor do they compare different models in terms of computational latency or resource usage. Moreover, the performance of larger-scale LLMs is not discussed, and no comparison with other multimodal fusion methods is provided, which limits the generalizability and comparability of the conclusions.

Other minor writing issues:

1. The paper "From Words to Worth: Newborn Article Impact Prediction with LLM" appears three times in the reference list; only the officially published version should be retained.

2. The use of "domain-specific" in the abstract is inaccurate. For scientific impact prediction, "domain" more naturally refers to academic disciplines rather than "scientific vs. other fields," and the current phrasing may cause semantic confusion.

3. The repeated use of "multi-scale, heterogeneous, and dynamic" in the abstract and introduction lacks precise definition or quantification, making the research problem description somewhat vague.

4. The division of factors into "scientific factors" and "non-scientific factors" is not rigorous. Both are related to research activities; a more accurate description would be "intrinsic factors" (within the paper) and "extrinsic factors" (external environment).

5. The phrase "multiple impact indicators" in the introduction is not specific; clarifying which indicators are included would improve the clarity of the abstract.

**Questions:**

1. In your experiments, only a single graph neural network (GNN) model was compared. Could you explain why more recent graph learning methods were not included? Compared to these latest models, what do you consider to be the main advantages of your framework?

2. Could you further clarify how your method differs from other recent approaches that integrate structural and textual information, and what improvements it brings in terms of mechanism or performance?

3. Would it be possible to provide more detailed ablation studies to verify the individual contributions of each component?

4. Could you include inference latency results for different models? Additionally, please consider providing clearer definitions for several concepts in the paper, such as "multi-scale" and "domain-specific."

Overall, I believe the first three points reflect serious deficiencies in the rigor of the original submission. Any additions made during the rebuttal may not be sufficient to enhance the rigor and scientific validity.

---

> ### Author Response · Authors · 2025-11-27
> **Response to Reviewer ZeSd – Part I**
>
> Thank you for your insightful feedback and concerns. Below, we provide a detailed response to address your comments and also we will incorporate these discussions and results into our revised paper.
>
> [**Question 1: There is limited task-specific mechanistic innovation or new theoretical insights for the impact prediction task.**]
>
> Thank you for raising this comment. We acknowledge that our work prioritizes practical solutions over theoretical novelty, and we appreciate the opportunity to clarify our contributions: (1) **LLM Integration Insight**: We find that directly fine-tuning LLMs induces systematic prediction bias severely, which means models overfit to training-specific values and underestimate citations (Section 5.2, Figure 3b). This negative result motivates our prefix-tuning design and may help future work avoid similar pitfalls. (2) **Multi-Scale Formulation**: We jointly model yearly and monthly forecasting, revealing that finer temporal granularity is easier to predict (Tables 3-4). This is a practical observation for researchers choosing prediction horizons. (3) **Contributed datasets and Empirical Findings**: Our ablations show reference links matter most, venue/institution provide complementary signals, and Mathematics-trained models transfer best across domains (Section 5.3). We agree these are primarily empirical contributions rather than theoretical advances. Our goal is to provide a solid benchmark and effective framework that lowers the barrier for future research. We hope the dataset, tools, and identified failure modes can serve as useful starting points for the community. We will clarify these contributions more explicitly in the revision.
>
> [**Question 2: The experiments didn’t cover the methods from 2023-2024, or the paper should explain the rationale for model selection or exclusion. Compared to these latest models, what do you consider to be the main advantages of your framework?**]
>
> We apologize for the confusion and thank you for raising this important question. Our work primarily explores impact prediction from an LLM perspective, so we focus on comparing LLM-based methods (including 2025 approaches) alongside traditional baselines (SciBERT, graph-based methods like HINTS). Regarding recent methods such as H2CGL (2023) and DPPDCC (2024): these approaches differ fundamentally in dataset construction. They use snapshot-based methods that store separate citation graphs at different time points, which presents two challenges for our setting: (1) this requires storing large citation graphs at multiple timestamps, and (2) more critically, it causes historical citation behavior leakage during training, conflicting with our newborn paper setting where we treat all papers as having no prior citations.  However, following the reviewer's suggestion, we adapted these methods (just remove the snapshot setting) to the newborn paper setting for fair comparison. Please see our general response.
>
>
> [**Question 3: The contributions of each module is not ablated. Also, the FasTtext temporal embedding should be evaluated separately.**]
>
> We thank the reviewer for this suggestion. We did not include a detailed ablation study in the original submission because removing individual modules results in configurations similar to existing baselines (e.g., text-only resembles SciBERT, graph-only resembles HINTS). We instead allocated space to cross-domain analysis and scientific factor exploration (Section 5.3). Following your suggestion, we conducted explicit ablation experiments to clarify each module's contribution as follows. The results confirm that each component contributes to the final performance. Notably, the prefix-tuning module provides larger gains than graph features alone, while text features show relatively smaller impact compared to graph and prefix components. This suggests that injecting structural signals into the LLM reasoning space via prefix tokens is more effective than simply combining features. We will include this ablation table in our revised paper.
>
> | Method | Year_1 | Year_2 | Year_3 | Year_4 | Year_5 | Avg |
> |--------|--------|--------|--------|--------|--------|------|
> | **Our** | **0.6772** | **0.6932** | **0.7809** | **0.8022** | **0.9177** | **0.7719** |
> | wo Prefix | 0.7387 | 0.8740 | 0.9584 | 1.0102 | 1.0698 | 0.9302 |
> | wo graph | 0.7021 | 0.8156 | 0.894 | 0.9521 | 0.9944 | 0.8716 |
> | wo time | 0.6782 | 0.6927 | 0.8011 | 0.8182 | 0.8904 | 0.7761 |
>
> **Table 1: Previous Test Set (Yearly)**
>
> | Method | Year_1 | Year_2 | Year_3 | Year_4 | Year_5 | Avg |
> |--------|--------|--------|--------|--------|--------|------|
> | **Our** | **0.7170** | **0.7426** | **0.7892** | **0.8302** | **0.9514** | **0.7745** |
> | wo Prefix | 0.7649 | 0.9084 | 1.0003 | 1.0470 | 1.0639 | 0.9569 |
> | wo graph | 0.7565 | 0.885 | 0.9637 | 1.0248 | 1.0705 | 0.9400 |
> | wo time | 0.7203 | 0.7499 | 0.8022 | 0.8339 | 0.9798 | 0.8172 |
>
> **Table 2: Fresh Test Set (Yearly)**

---

> > ### Author Response · Authors · 2025-11-27
> > **Response to Reviewer ZeSd – Part II**
> >
> > [**Question 4: The significance testing for results and computational latency of different methods are not provided. Also, the larger LLMs and other multimodal fusion methods are not provided.**]
> >
> > Thanks for this comment. Regarding significance testing, we conducted experiments with 5 random seeds. The standard deviation of average RMSLE is ±0.0098 (Previous) and ±0.018 (Fresh), confirming the stability of our results and the significance of reported improvements.
> >
> > We also provide the training, and inference time as follows. As shown, graph-based methods are most efficient. However, our method maintains competitive efficiency compared to other LLM-based approaches (NAIP) while achieving substantially better performance.
> >
> > | Method | Train (hour) | Inference (ms per case) |
> > |--------|--------------|-------------------------|
> > | Scibert | 2.90 (100% data, 6 epoch) | 2.50ms |
> > | HINTS | 6.48 (100% data, 6 epoch) | 0.98ms |
> > | NAIP | 8.92 (10% data, 4 epoch) | 324ms |
> > | **Ours** | **13.68 (100% data, 6 epoch)** | **14.83ms** |
> >
> > Regarding the selection of only small LLMs, the reason is that our dataset contains ~200K training samples. We intentionally chose smaller LLMs (≤8B) for prefix-tuning to balance performance and efficiency. As shown in Table 3, scaling from Qwen-4B to Qwen-8B actually degrades performance (1.33 → 2.04 avg. RMSLE), suggesting that larger models are not necessarily beneficial for this task and may introduce overfitting. We believe exploring efficient adaptation of larger LLMs is valuable future work.
> >
> > Regarding multimodal fusion methods,our current framework does not incorporate multimodal content (e.g., figures, tables) as input features. If we misunderstood your question, we would be happy to clarify further.
> >
> > [**Question 5: The definition of "domain-specific", “multi-scale, heterogeneous, and dynamic”.**]
> >
> > We apologize for the confusion of these definitions. We clarify these terms below:
> >
> > * **Domain-specific**: "Domain" refers to scientific disciplines (e.g., Physics, Mathematics, Biology), as illustrated in Figure 5(a). Different domains exhibit distinct citation patterns and dynamics, and also require LLMs to understand different domain knowledge.
> >
> > * **Multi-scale**: This refers to our prediction across different temporal windows, both yearly (1-5 years) and monthly (1-5 months) horizons, as shown in Tables 3 and 4.
> >
> > * **Heterogeneous**: This describes the diverse types of features and relations in our framework, including different node types (papers, authors, venues, institutions) and edge types (citations, authorship, affiliation), as defined in Section 4.1.
> >
> > * **Dynamic**: This captures the evolving nature of citation behavior that a paper's impact changes over time, and citation counts vary across different temporal windows.
> >
> > We will add clearer definitions of these terms in the revision to avoid ambiguity. Thank you for pointing this out.
> >
> > [**Question 6: Could you further clarify how your method differs from other recent approaches that integrate structural and textual information, and what improvements it brings in terms of mechanism or performance?**]
> >
> > Thanks for your constructive question. We agree that structural and non-structural integration frameworks has done some similar exploration, such as concatenating LLMs summarization as additional features. The value of our work lies mainly in two aspects: (1) **Task-specific requirements**: Scientific impact prediction differs from general structure-text integration tasks. Predicting a paper's future citations requires understanding domain-specific factors: research novelty, methodological rigor, topical relevance, and field-specific citation norms. These factors are implicit in text and cannot be captured by simple text encoders, and they require deeper semantic reasoning. (2) **Graph-feature as tokens**: General approaches typically use text encoders (e.g., BERT) as static feature extractors, which capture surface-level semantics but lack domain reasoning capability. In contrast, LLMs pretrained on scientific corpora encode rich domain knowledge about research quality, topic importance, and scholarly conventions. Our prefix-tuning strategy injects graph-derived structural signals (author reputation, venue prestige, citation networks) into the LLM, enabling it to reason about impact by jointly considering "what the paper says" and "who wrote it, where it was published, and what it builds upon." This design leverages LLMs' domain knowledge for scientific understanding rather than treating them as generic encoders.
> >
> > [**Question 7: Minor issue of paper citation format.**]
> >
> > Thanks. We’ll carefully fix them in our revised version.
> >
> > We hope these clarifications will address your concern and not overshadow the contribution of our work. Please feel free to leave any questions that need to be more detailed and explained.

---

### Author Response · Authors · 2025-11-27
**General response to all reviewers**

Thanks to all reviewers’ time and effort in reviewing our paper. Below, we first address a common concern regarding additional baselines. Following the reviewers' suggestions, we incorporated H2CGL and DPPDCC by adapting them to our newborn paper setting (removing the snapshot mechanism to prevent citation leakage). The results are shown below. Our method maintains consistent advantages over these approaches. We will include these comparisons and clarify our baseline selection rationale in the revision.

| Method | Year_1 | Year_2 | Year_3 | Year_4 | Year_5 | Avg |
|--------|--------|--------|--------|--------|--------|------|
| **Our** | **0.6772** | **0.6932** | **0.7809** | **0.8022** | **0.9177** | **0.7719** |
| H2CGL[1] | 0.7965 | 0.9250 | 1.0037 | 1.0648 | 1.1105 | 0.9800 |
| DPPDCC[2] | 0.7764 | 0.9051 | 0.9866 | 1.0485 | 1.0949 | 0.9622 |

**Table 1: Previous Test Set (Yearly)**

| Method | Year_1 | Year_2 | Year_3 | Year_4 | Year_5 | Avg |
|--------|--------|--------|--------|--------|--------|------|
| **Our** | **0.7170** | **0.7426** | **0.7892** | **0.8302** | **0.9514** | **0.7745** |
| H2CGL[1] | 0.8585 | 0.9995 | 1.0987 | 1.1431 | 1.1558 | 1.0511 |
| DPPDCC[2] | 0.8209 | 0.9636 | 1.0600 | 1.1061 | 1.1233 | 1.0147 |

**Table 2: Fresh Test Set (Yearly)**

[1] H2CGL: Modeling dynamics of citation network for impact prediction, Information Processing and Management, 2023.

[2] Predicting Scientific Impact Through Diffusion, Conformity, and Contribution Disentanglement, CIKM 2024.

---

### Author Response · Authors · 2025-12-03
**Summary of Rebuttal**

Dear ACs and SACs,

We understand that the meta-reviewing is over workload this year, so we provide a brief summary of our rebuttal responses.

We thank all reviewers for their efforts in reviewing our paper and constructive comments. We are encouraged that the importance and value of our focused task were recognized by Reviewer **ZeSd (Score: 2)**, **EkwV (Score: 4)** and **PPab (Score: 6)**. Also, our main contribution of integrating graph networks into LLMs, the contributed large-scale datasets are acknowledged by all Reviewer **ZeSd (Score: 2)**, **EkwV (Score: 4)**, **uMUB (Score: 6)**, **PPab (Score: 6)**, and also the effectiveness of our method  by Reviewer **EkwV (Score: 4)**, **uMUB (Score: 6)**, **PPab (Score: 6)**.

During the rebuttal, we made the following efforts to address reviewers’ comments:

* More baseline comparison. (**ZeSd**, **EkwV**, **PPab**)
  * We focused primarily on LLM-based methods in the main paper, but we have now additionally implemented two graph-based baselines (DPPDCC, 2024 and H2CGL, 2023) to further support the effectiveness of our approach.
  * We also conducted a significance test using five random seeds and provided variance results to strengthen the reliability of our improvements. (**ZeSd**, **uMUB**)

* Conduct additional ablation studies. (**ZeSd**)
  * We performed more extensive ablation experiments, removing each module individually to demonstrate the contribution of each component.

* Clarifications of key concepts and our main contributions.
  * We clarified how our work differs from existing methods and emphasized both its effectiveness and its task-specific requirements.  (**ZeSd**, **EkwV**)
  * We acknowledged that our focus is primarily on addressing a practical application challenge rather than advancing theoretical insights; nonetheless, we believe our work offers meaningful contributions that help broaden the scientific impact of modeling approaches. (**ZeSd**)
  * We also clarified the concepts of domain-specific, multi-scale, heterogeneous, and dynamic data, and described the dataset distribution in detail.  (**ZeSd**, **EkwV**)

* Explanation regarding the use of smaller LLMs.
  * We explained that the task is computationally intensive and our dataset contains over 200k cases with large citation graphs. For this reason, smaller LLMs offer a more appropriate balance between effectiveness and efficiency.  (**ZeSd**, **PPab**)

* Explain some misunderstandings.
  * We clarified that our dataset does not include multimodal data, and therefore multimodal models were not included. (**ZeSd**)
  * We also explained that the main advantage of our method lies in its integration of graph features, which cannot be achieved solely through fine-tuning with hyperparameter optimization. This point addresses the misunderstanding that led Reviewer **EkwV** to temporarily keep their score unchanged before the comment window closed.  (**EkwV**)

We hope that these clarifications and additional experiments fully address the concerns raised by Reviewer **ZeSd (Score: 2)**. Moreover, their comments were genuinely helpful in improving the clarity, positioning, and overall quality of our work.

Best regards,

The authors.

---

### Meta-Review · Area_Chair_f47A · 2026-01-06

**Summary:**

This method employs a heterogeneous graph neural network to model an academic network comprising multiple node types, including papers, authors, institutions, and venues, and maps the resulting structural representations to learnable prefix tokens within the prefix-tuning framework. The initial score was 2/4/6/6, and the score remained unchanged before the reset.

The reviewers’ concerns mainly focused on: 1) lacking deep, task-specific mechanistic innovation (Reviewer ZeSd); 2) insufficient comparisons against baselines (Reviewers ZeSd, EkwV, PPab); 3) insufficient analyses on the model designs (Reviewers ZeSd, EkwV) and et al. According to the authors’ rebuttal, Reviewer EkwV remained unconvinced of the submission’s technical novelty and maintained the score of 4, a concern also shared by Reviewer ZeSd.

After a careful assessment of the submission, reviews, response, and discussion, the AC recommends rejection. The authors are encouraged to revise and refine the manuscript in accordance with the reviewers’ feedback for a future submission.

**Reviewer Concerns:**

The reviewers’ concerns mainly focused on: 1) lacking deep, task-specific mechanistic innovation (Reviewer ZeSd); 2) insufficient comparisons against baselines (Reviewers ZeSd, EkwV, PPab); 3) insufficient analyses on the model designs (Reviewers ZeSd, EkwV) and et al. According to the authors’ rebuttal, Reviewer EkwV remained unconvinced of the submission’s technical novelty and maintained the score of 4, a concern also shared by Reviewer ZeSd.

**Reviewer Scores:**

The manuscript received initial review scores of 2/4/6/6. After the rebuttal/discussion and before the reset, the score remained unchanged.

Since several concerns raised by the reviewers may remain unresolved after the rebuttal (see 'Reviewer Concerns'), I would approximate 2/4/6/6 as the final score.

---

### Decision · Program_Chairs · 2026-01-26

Reject